# Rapid Prediction of Two-Dimensional Airflow in an Operating Room using Scientific Machine Learning

**Gary L. Collins**[*], **Alexander New**[*], **Ryan A. Darragh**, **Brian E. Damit**, **Christopher D. Stiles**
Research and Exploratory Development Department
Johns Hopkins University Applied Physics Laboratory
Laurel, Maryland 21044
{gary.collins, alex.new, ryan.darragh, brian.damit, christopher.stiles}@jhuapl.edu
[*] equal contribution

## Abstract

We consider the problem of using scientific machine learning (SciML) to rapidly predict solutions to systems of nonlinear partial differential equations (PDEs) defined over complex geometries. In particular, we focus on modeling how airflow in operating rooms (ORs) is affected as the position of an object within the OR varies. We develop data-driven and physics-informed operator-learning models based on the deep operator network (DeepONet) architecture. The DeepONet models are able to accurately and rapidly predict airflow solutions to novel parameter configurations, and they surpass the accuracy of a random forest (RF) baseline. Interestingly, we find that physics-informed regularization (PIR) does not enhance model accuracy, partially because of misspecification of the physical prior compared to the data's governing equations. Existing SciML models struggle in predicting flow when complex geometries determine localized behavior.

## 1 Introduction

Many problems of engineering interest can be solved by modeling a system of partial differential equations (PDEs) across a non-uniform spatiotemporal domain, such as weather forecasting [1], blood cell modeling [2], materials science [3], cellular signaling [4], and hypersonics [5]. Two key challenges typically encountered are that (i) solving the relevant PDE systems requires large amounts of time and computation, and (ii) the mathematical model specified by the PDE may not be fully representative of the underlying physical phenomena. Increasingly, these problems have been resolved with the use of scientific machine learning (SciML) [6, 7, 8].

Here, we focus on the use of SciML for the task of rapidly predicting airflow in operating rooms (ORs), in particular, modeling how airflow changes as the placement of objects within the OR changes. This is a relevant and interesting challenge for SciML for a number of reasons. Unidirectional flow (UDF) is often used in ORs to attempt to reduce the occurrence of surgical site infections (SSIs) by continuously supplying the surgical zone with clean air and washing away potentially pathogen-laden particles [9, 10]. However, the interaction of healthcare workers and objects, such as surgical lights and surgical tables, with the airflow may be difficult to predict and sensitive to the specific geometry and flow conditions in the room which may create recirculation regions that potentially allow aerosols to enter the surgical zone [11, 12, 13, 14, 15, 16, 17].

By developing a method to quickly predict the resulting airflow in an OR given a configuration of medical equipment, the effectiveness of UDF and other ventilation strategies can more easily be determined. Understanding the effects of object configuration inside an OR is important for reducing negative health outcomes; however, traditional computational fluid dynamics (CFD) methods for evaluating a flow field are computationally costly and impractical to perform in large numbers.

NeurIPS 2023 AI for Science Workshop.

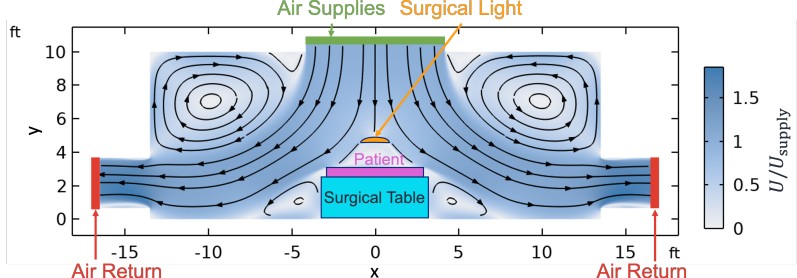

Figure 1: We solve airflows over this 2D-OR geometry and surgical light placement. The position of the light varies based on the OR configuration parameters $\psi$.

In this paper, we use SciML to develop three models (Sections 2.1 and 2.2) based on the deep operator network (DeepONet) [18] operator-learning architecture, that predict the velocity and pressure fields in a two-dimensional representation of an operating room (2D-OR) with respect to the placement and orientation of a single surgical light (Appendix B). The three models include a purely data-driven model and two physics-informed models. Once trained, the SciML surrogates are significantly more computationally efficient at developing solutions than traditional CFD while generalizing accurately across the parameter space (Section 3.1). In addition, the DeepONet models outperform a random forest baseline model by an order of magnitude. For some cases, the SciML models struggle to capture the flow behavior in the wake of the surgical light, and physics-informed approaches have marginal losses in accuracy compared to the purely data-driven model, which highlights the difficulty in capturing complex physical phenomena (Section 3.2). Although fidelity is lost when using a two-dimensional (2D) OR geometry as opposed to a three-dimensional (3D) one, the selection of a 2D-OR geometry for this work was made in order to increase the number of ground truth simulations available for training [19]. Extensions to this work include evaluation on three-dimensional geometries, as well as optimization of geometry configurations to satisfy target criteria.

## 2 Methods

### 2.1 Problem setup

We consider a 2D representation of an OR-like geometry (Figure 1) with a surgical table, air supplies and returns, and a light parameterized by $\psi = (x_{\text{light}}, y_{\text{light}}, \theta_{\text{light}})$, where $(x_{\text{light}}, y_{\text{light}})$ is the position of the center of the light and $\theta_{\text{light}}$ the angle the bottom face of the light makes with the floor. For a given $\psi$, $\Omega(\psi)$ denotes the set of points $\mathbf{x} = (x, y)$ in this OR. Further details are in Appendix B. Airflow in this domain is treated as incompressible and is described by the field $\mathbf{U} = (u, v, p)$, where $u$ and $v$ are $x$- and $y$- component velocities, and $p$ is pressure. These variables obey the incompressible steady-state Navier-Stokes equations (NSE):

$$\mathcal{N}_c(\mathbf{U}) = \partial_x u + \partial_y v = 0 \quad \text{(continuity)} \tag{1}$$

$$\mathcal{N}_x(\mathbf{U}) = u\,\partial_x u + v\,\partial_y u + \frac{1}{\rho}\partial_x p - \nu\Delta u = 0 \quad (x\text{-momentum conservation}) \tag{2}$$

$$\mathcal{N}_y(\mathbf{U}) = u\,\partial_x v + v\,\partial_y v + \frac{1}{\rho}\partial_y p - \nu\Delta v = 0 \quad (y\text{-momentum conservation}), \tag{3}$$

where $\rho$ is the density of air (here $1.2043\,\text{kg/m}^3$), and $\nu$ is the kinematic viscosity of air (here $1.5062 \times 10^{-5}\,\text{m}^2/\text{s}$). In addition, by taking the divergence of the momentum equations and applying the continuity equation to simplify, we derive an additional Poisson equation constraint for pressure:

$$\mathcal{N}_p(\mathbf{U}) = \frac{1}{\rho}\Delta p + (\partial_x u)^2 + 2\,\partial_y u\,\partial_x v + (\partial_y v)^2 = 0 \quad \text{(Poisson pressure constraint).} \tag{4}$$

The CFD-based ground-truth data is generated using Reynolds-averaged Navier-Stokes (RANS) simulations (specifically, the realizable $k-\varepsilon$ model [20]), which introduce additional turbulence variables and equations. Although here we consider only fluid flow models, additional quantities, such as airborne particle distributions, can be coupled to the fluid flow equations [21].

## 2.2 Operator models

We use a DeepONet [18]-type architecture $\hat{\mathbf{U}}^{\text{norm}} : \mathbb{R}^2 \times \mathbb{R}^3 \rightarrow \mathbb{R}^3$ to predict the normalized state $\mathbf{U}^{\text{norm}} = (u^{\text{norm}}, v^{\text{norm}}, p^{\text{norm}})$ at a position $\mathbf{x} \in \Omega(\psi)$ in the 2D-OR domain:

$$\hat{\mathbf{U}}^{\text{norm}}(\mathbf{x}, \psi) = (\hat{u}^{\text{norm}}, \hat{v}^{\text{norm}}, \hat{p}^{\text{norm}})(\mathbf{x}, \psi) = \begin{cases} f_{(\text{combine},u)}(f_{\text{trunk}}(\mathbf{x}) \odot f_{\text{branch}}(\psi)) \\ f_{(\text{combine},v)}(f_{\text{trunk}}(\mathbf{x}) \odot f_{\text{branch}}(\psi)) \,, \\ f_{(\text{combine},p)}(f_{\text{trunk}}(\mathbf{x}) \odot f_{\text{branch}}(\psi)) \end{cases} \quad (5)$$

where $f_{(\text{combine},\cdot)}$, $f_{\text{trunk}}$, and $f_{\text{branch}}$ are neural networks (NNs), and $\odot$ is element-wise multiplication. We combine spatial and parameter decoders $f_{\text{trunk}}$ and $f_{\text{branch}}$ with joint encoders $f_{(\text{combine},\cdot)}$ to increase expressivity of the network, similar to other recent operator-learning research [22].

We seek a model $\hat{\mathbf{U}}^{\text{norm}}$ that can generalize over the space of configuration parameters $\psi$. Thus, we sample $N$ parameter values $\psi_n$, $n = 1, \ldots, N$ to obtain $N$ simulations $\mathcal{D}(\psi_n) = \{(\mathbf{x}_{n,m}, \mathbf{U}_{n,m})\}_{m=1}^{M_n}$ consisting of field values $\mathbf{U}_{n,m} \in \mathbb{R}^3$ observed at mesh points $\mathbf{x}_{n,m} \in \Omega(\psi_n)$, where $M_n$ is the number of mesh points for simulation $n$.

NNs are trained on normalized data, but the NSE are defined in physically-meaningful units. Thus, $f_{\text{trunk}}$ and $f_\psi$ are multi-layer perceptrons (MLPs) with normalization layers:

$$f_{\text{trunk}}(\mathbf{x}) = \text{MLP}\left(\frac{\mathbf{x} - \boldsymbol{\mu}_{\mathbf{x}}}{\boldsymbol{\sigma}_{\mathbf{x}}}\right), \quad (6)$$

where $\boldsymbol{\mu}_{\mathbf{X}}$ and $\boldsymbol{\sigma}_{\mathbf{X}}$ are the mean and standard deviation of $\mathbf{x}$ across the training set, the division is carried out element-wise, and $f_{\text{branch}}$ is defined similarly. Further, the normalized output of the model $\hat{\mathbf{U}}^{\text{norm}}$ is subsequently normalized to produce physical quantities:

$$\hat{\mathbf{U}} = \boldsymbol{\sigma}_{\mathbf{U}} \odot \hat{\mathbf{U}}^{\text{norm}} + \boldsymbol{\mu}_{\mathbf{U}}, \quad (7)$$

where $\boldsymbol{\mu}_{\mathbf{U}}$ and $\boldsymbol{\sigma}_{\mathbf{U}}$ are the mean and standard deviation of $\mathbf{U}$ across the training set.

The DeepONet $\hat{\mathbf{U}}$ is trained by minimizing a loss function defined from both data-based error and physics-informed regularization (PIR) (Eqs. 1, 2,3,4). This yields the loss function $\mathcal{L}$, minimized with stochastic gradient descent (SGD) and Adam [23]:

$$\begin{aligned} \mathcal{L} = \;& \frac{1}{N}\sum_{n=1}^{N}\frac{1}{M_n}\sum_{m=1}^{M_n}\frac{1}{3}\left\|\mathbf{U}_{n,m}^{\text{norm}} - \hat{\mathbf{U}}^{\text{norm}}(\mathbf{x}_{n,m}, \psi_n)\right\|_1 + \\ & \frac{1}{N}\sum_{n=1}^{N}\frac{1}{M_n}\sum_{m=1}^{M_n}\left(\lambda_x|\mathcal{N}_x(\hat{\mathbf{U}})(\mathbf{x}_{n,m}, \psi_n)| + \lambda_y|\mathcal{N}_y(\hat{\mathbf{U}})(\mathbf{x}_{n,m}, \psi_n)|\right) + \\ & \frac{1}{N}\sum_{n=1}^{N}\frac{1}{M_n}\sum_{m=1}^{M_n}\left(\lambda_c|\mathcal{N}_c(\hat{\mathbf{U}})(\mathbf{x}_{n,m}, \psi_n)| + \lambda_p|\mathcal{N}_p(\hat{\mathbf{U}})(\mathbf{x}_{n,m}, \psi_n)|\right), \end{aligned} \quad (8)$$

where $|\mathcal{N}_x(\hat{\mathbf{U}})|$, $|\mathcal{N}_y(\hat{\mathbf{U}})|$, $|\mathcal{N}_c(\hat{\mathbf{U}})|$, and $|\mathcal{N}_p(\hat{\mathbf{U}})|$ are physics-informed residuals, and $\lambda_x, \lambda_y, \lambda_c, \lambda_p > 0$ are regularization strengths.

Here, each $\lambda_x, \lambda_y, \lambda_c$, and $\lambda_p$ is based on a shared $\lambda_0 = 1$. Specifically, $\lambda_0$ is scaled based on characteristic length $L$ and velocity $U$. This scaling makes each PIR loss component dimensionless:

$$\lambda_c = \lambda_0 \frac{L}{V}, \quad (9)$$

$$\lambda_x = \lambda_y = \lambda_0 \frac{L}{V^2}, \quad (10)$$

$$\lambda_p = \lambda_0 \frac{L^2}{V^2}, \quad (11)$$

where the length $L$ (0.75 ft) and velocity $V$ (35 ft/min) scales are the length of the major semi-axis of the surgical light and inlet velocity magnitude, respectively.

## 2.3 Implementation

We implement and train NNs with `jax` [24], `flax` [25], and `optax` [26]. Derivatives in PIR terms are calculated with forward-mode automatic differentiation (AD) [27], with nested forward-over-forward AD for second-order derivatives. Gradients with respect to NN weights are calculated with reverse-mode AD. The computational cost of nested AD scales exponentially with the order of the derivative. For example, see [28], which reports, in `jax`, roughly an order of magnitude more time needed to evaluate the second derivatives of an MLP's prediction with respect to its inputs, than just the prediction directly. This could be mitigated by the use of techniques like Taylor-mode AD [29]. In practice, we use a "warm-start" strategy and turn off regularization ($\lambda = 0$) for the first 5000 epochs of training to obtain a partially converged model.

PIR constitutes an over-determined set of equations, in that the Poisson pressure constraint (Eq. 4) is derived from the NSE (Eqs. 1, 2, 3). Thus, if the Poisson pressure constraint is satisfied ($\mathcal{N}_p(\mathbf{U}) = 0$), the continuity requirement is also satisfied ($\mathcal{N}_c(\mathbf{U}) = 0$). Because the continuity residual $\mathcal{N}_c$ is simpler to calculate than the Poisson pressure constraint $\mathcal{N}_p$, we include both components in $\mathcal{L}$. Furthermore, the derivatives in the continuity equation also occur in other equations, so there is negligible additional incurred computational cost for calculating continuity.

In addition, we do not expect to be able to exactly satisfy every term in the loss function (Eq. 8), because the bias introduced by our regularization is misspecified compared to the data-generating process. That is, the ground truth data was generated using a turbulence model, and this is not included in our PIR. We use this form of PIR as an approximate proxy for the turbulence models CFD simulations use in order to evaluate the effectiveness of physical priors with reduced complexity.

## 3 Results

The set of steady-state solutions are partitioned with 1000 light orientations for training, 200 light orientations for validation, and the remaining 107 light orientations for testing. We evaluate trained models on a simulation $n$ with the relative error (Eq. 12):

$$E_{\text{rel}}(\boldsymbol{\psi}_n) = \frac{\sqrt{\sum_{m=1}^{M_n} ||\mathbf{U}_{n,m}^{\text{norm}} - \hat{\mathbf{U}}^{\text{norm}}(\mathbf{x}_{n,m}, \boldsymbol{\psi}_n)||^2}}{\sqrt{\sum_{m=1}^{M_n} ||\mathbf{U}_{n,m}^{\text{norm}}||^2}} \tag{12}$$

See Table 4 in Appendix C for details on the hyperparameters used to train networks.

### 3.1 Model evaluation

We first assessed baseline performance on this dataset with a random forest (RF) [30], using SCIKIT-LEARN's [31] RANDOMFORESTREGRESSOR function. After hyperparameter selection on the validation set, when evaluated on the same test set of 107 light orientations, the RF achieved a mean relative error of 0.21. The distribution of test set relative errors is given in Figure 7 in Appendix D, and the hyperparameter selection procedure is in Table 3 in Appendix C. Due to the complexity of the 2D-OR geometry, some popular SciML-based methods for solving PDEs, such as the Fourier neural operator (FNO) [32], are not immediately applicable. However, MLP-based DeepONet architecture could be strengthened by adapting ideas from other approaches, such as graph networks [33].

In Figure 2, we plot the data-based error and PIR terms, for both the training and validation splits. The models with PIR are initialized from the unregularized model after 5000 epochs and run for 2000 epochs, while the unregularized model also trained for an additional 2000 epochs. The unregularized DeepONet initially improves in both training and validation errors but reaches a minimum validation data error by 700 epochs. The parameters of the unregularized model at 5000 epochs of training are used to initialize two PIR models: one regularized by the continuity equation (PIR-continuity) and one regularized by the NSE and Poisson pressure equation (PIR-all). After 2000 additional epochs, enforcing PIR in the loss causes each model's physical residuals to decrease by a factor of two; enforcing PIR does not substantively affect the validation set's data loss. At the final epoch for all three models, the majority of the relative data errors are below $10\%$ as shown in Figure 6 in Appendix D. Further, the mean relative errors are also low for each partition, as shown by Table 1. This is significantly better than the random forest baseline's mean relative error of 0.21. In the 2D-OR

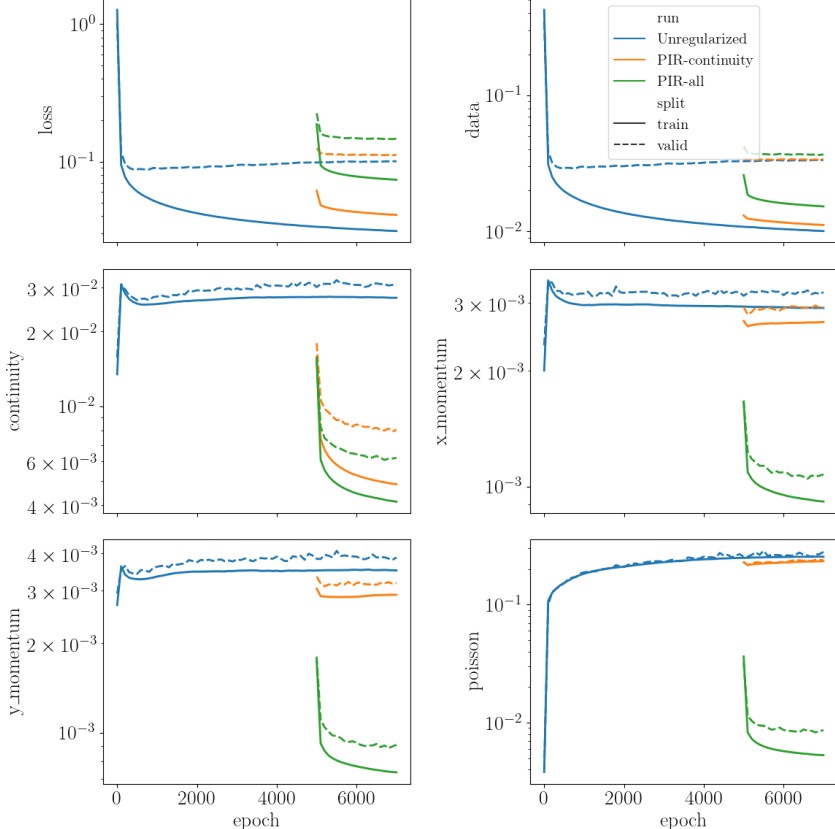

Figure 2: The loss function components, for training and validation splits, from training the Deep-ONets on the 2D-OR data. PIR models were initialized from the unregularized model's state after 5000 epochs and then trained for 2000 epochs. Enforcing PIR decreases the physics-based residuals but does not significantly affect data error for the validation set.

Table 1: Mean relative errors of all models for each data partition after 7000 epochs of training.

| Dataset | Unregularized | PIR-continuity | PIR-all | RF |
|---|---|---|---|---|
| Training | 0.0247 | 0.0338 | 0.0490 | 0.0206 |
| Validation | 0.0625 | 0.0670 | 0.0757 | 0.2031 |
| Testing | 0.0669 | 0.0707 | 0.0786 | 0.2001 |

geometry, mesh points are more highly concentrated near the bed and light, because a higher spatial resolution is needed to accurately resolve the fluid behavior. This means that the relative error is more affected by prediction errors in highly-sampled geometry regions, compared to less-sampled regions. If we interpolate our CFD ground truth onto a regular mesh and evaluate the trained models on that mesh, the effect of higher mesh concentration is removed. As seen in Figure 10 in Appendix D, this results in all models having relatively similar distributions of error. Thus, PIR mostly affects the regions near geometries (i.e., the surgical table and light).

We can also characterize this in terms of how much the regularized models' predictions change compared to the unregularized model. For this, we compute an average change:

$$\overline{|\Delta \mathbf{U}(\boldsymbol{x})|} = \frac{1}{N} \sum_{n=1}^{N} \left\| \hat{\mathbf{U}}_{\text{Unregularized}}^{\text{norm}}(\mathbf{x}, \boldsymbol{\psi}_n) - \hat{\mathbf{U}}_{\text{PIR}}^{\text{norm}}(\mathbf{x}, \boldsymbol{\psi}_n) \right\|, \tag{13}$$

where $\hat{\mathbf{U}}_{\text{Unregularized}}^{\text{norm}}$ is the unregularized DeepONet model at the 5000th epoch and $\hat{\mathbf{U}}_{\text{PIR}}^{\text{norm}}$ is a PIR model after 2000 additional epochs. In Figure 11 in Appendix D, we show that most of the changes are concentrated in the region between the light and the surgical table and near the room boundary.

However, given that the loss functions lack boundary conditions, the effects of the interaction of the fluid and geometries are absent in PIR and must arise from the data-loss alone. In regions nearer to geometries, where flow field tends to have larger gradients and where the influence of physics is not captured by the PIR losses (e.g. boundary conditions and turbulence), the PIR models are detrimental.

Compared to relative error, PIR has the weakness that its minimizing solutions are not unique. A zero-field satisfies the NSE; thus, PIR alone is not sufficient to yield accurate flow solutions. However, relative error lacks the physics-based grounding of PIR, and so tracking both data-based and physics-informed errors for SciML models provides additional metrics for interrogation and understanding. Due to having lower physics-based error, models trained with PIR may be superior in some application settings. Furthermore, future work could consider loss functions that adaptively trade between physics- and data-based error depending on the location in the domain, such as prioritizing physics-based error in domain interiors and data-based error near boundaries.

## 3.2 Analysis of results

Fluid flowing around an object may separate into a wake that is of high interest to researchers, but is difficult to accurately model. Within the healthcare domain, the wake of a piece of equipment in the UDF region may have a direct effect on the potential exposure of patients to potentially pathogen-laden particles as it will disrupt the UDF in regions directly above or on top of a patient.

Each model has its maximum relative error on the same configuration, shown in Figure 3; this configuration was in the validation split. A majority of the error appears in the wake of the light (i.e., the trapezoidal area under the light that extends to the surgical table), where alternating eddies appear in the horizontal velocity and pressure (vortex shedding), and the vertical velocity appears to turn upward (reverse flow). Vortex shedding and reverse flow are caused by viscous interaction between the fluid and the surface of an object (i.e., a boundary condition). Since boundary conditions are not explicitly represented in Eq. 8, PIR cannot improve a model's ability to predict them. Instead, whether or not a model predicts them will depend on its training data. In particular, the two nearest training configurations (with respect to Euclidean distance in the normalized parameter space) do not have vortex shedding or reverse flow, shown in Figure 8 (in Appendix D).

In contrast, for the validation case shown in Figure 4, the DeepONets accurately predict vortex shedding and reverse flow. For this configuration, the closest training cases have significant wake behaviors (Figure 9 in Appendix D). This supports the hypothesis of model reliance on training data. Examining the error across all cases reveals that the majority of the error for the DeepONets occur in the wake region. For each dataset partition, roughly 20 percent of the cases have a mean positive $v$-component of velocity in the wake of the light, indicating reverse flow. Table 2 shows that the DeepONets have more error in the wake of the light than outside of it; further, all models have significantly higher error in validation and testing cases with reverse flow than validation and testing cases without reverse flow. However, this discrepancy does not exist for the training cases, showing the dependency of data-driven losses for reverse flow prediction. Overall, this is indicative of the difficulty the DeepONets have in predicting complex fluid dynamics behaviors, such as vortex shedding and reverse flow, without sufficient data or codified physics.

Table 2: The DeepONets achieve the most error in predicting airflow in the domain's subregion corresponding to the wake of the OR's light, especially when reverse flow occurs. Specifically, we partition the 2D-OR into "within the wake of the light" and "outside the wake of the light" regions and calculate mean relative error (Eq. 12) within those regions for each model, for the test set.

| Region | Unregularized | PIR-continuity | PIR-all |
|---|---|---|---|
| Within the wake of the light, reverse flow | 0.1530 | 0.1515 | 0.1461 |
| Within the wake of the light, no reverse flow | 0.0974 | 0.0965 | 0.0978 |
| Outside the wake of the light | 0.0460 | 0.0498 | 0.0554 |

## 3.3 Computational cost

Using a trained DeepONet to predict a solution is substantially faster than traditional CFD. The average cost for a SciML model to generate a whole solution to a given light orientation is $\mathcal{O}(0.1)$ CPU-seconds. In contrast, the traditional CFD approach required an average of 52 CPU-minutes.

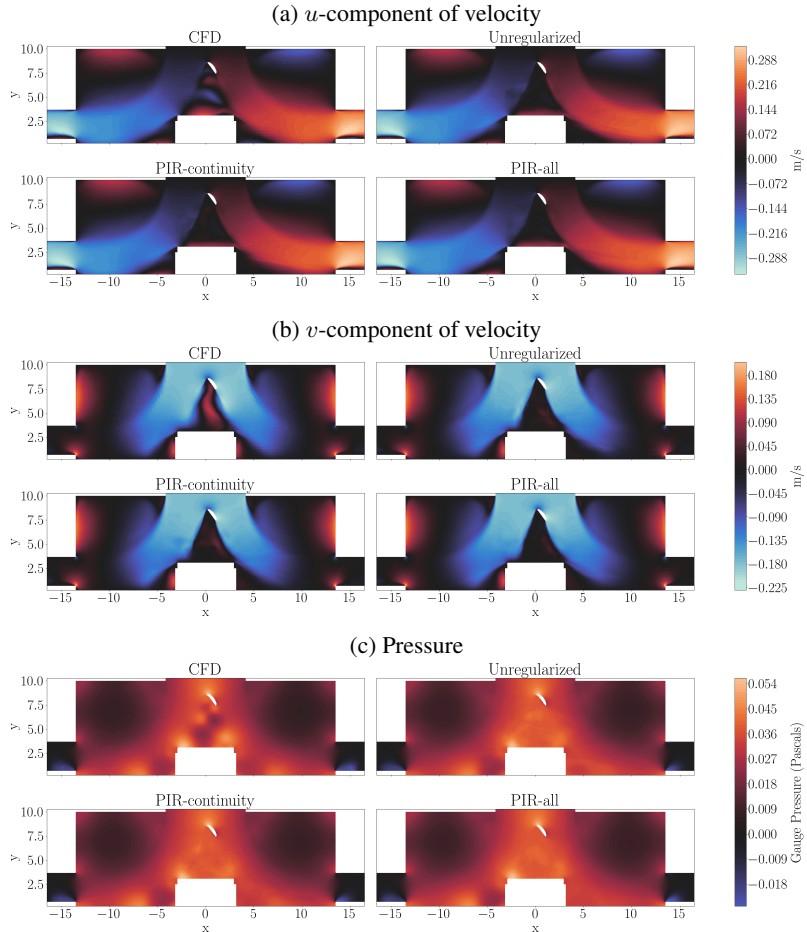

Figure 3: State predictions of each model on their highest error validation case, where the models fail to predict vortex shedding. For each model, the majority of the error occurs in the wake of the light. This configuration's light is $0.6\,\text{ft}$ to the right of the center, $8\,\text{ft}$ above the floor, and rotated $-54°$.

However, training cost is significant, especially using PIR. Using a NVIDIA A100 GPU, the unregularized DeepONet required 17.6 GPU-hours to complete 7000 training epochs. For the addditional 2000 epochs, the PIR-continuity model required an additional 19 GPU-hours, and the PIR-all model required an additional 117.4 GPU-hours. For predicting solutions, the time cost of unregularized and PIR models are equivalent, because their network architectures are the same.

In settings where only a small number of CFD simulations are required, the computational cost of training an DeepONet outweighs the cost of performing individual simulations. However, solving a design problem can require many simulations as parameter space is explored. Thus, having access to a trained SciML model can be efficient in this setting. Furthermore, DeepONet training time would be reduced if the amount of required training data could be reduced while still maintaining predictive performance. Such reduction has been shown to be possible in settings like materials science [34], and PIR could prove helpful in maximizing data efficiency.

## 4   Conclusion

DeepONets learned the influence of the placement and orientation of a surgical light inside an OR with UDF ventilation over a surgical table. Training, validation, and testing solutions were obtained using RANS. Three DeepONet models were produced: one arising from velocity and pressure losses and two that were warm-started from the previous model and then trained with PIR. The two PIR models used a continuity loss term and a weighted sum of continuity, momentum-conservation, and

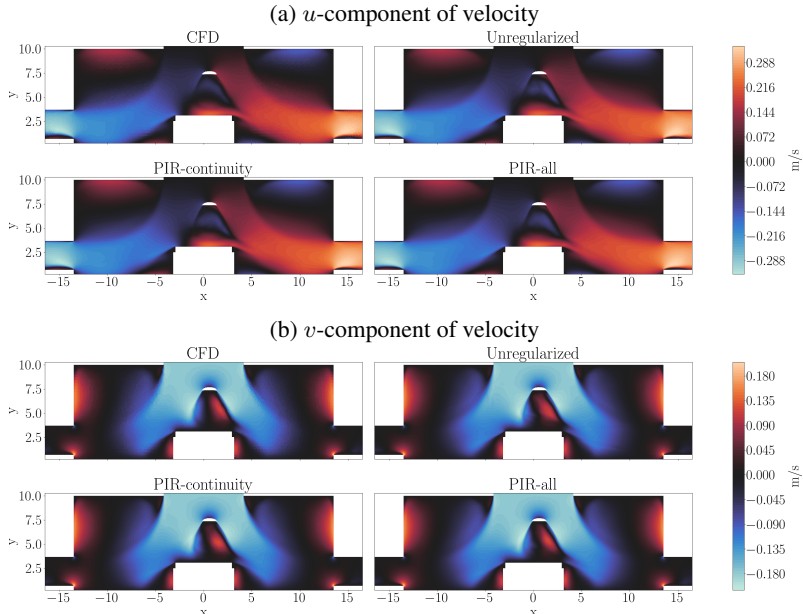

Figure 4: The velocity profiles of a validation case where the DeepONets accurately predict vortex shedding and reserve flow. The light orientation is $0.6\,\text{ft}$ to the right of center, $7.36\,\text{ft}$ above the ground, and facing directly downward ($0°$).

Poisson pressure loss terms. Although this particular problem is healthcare-focused, it reveals many insights on the application of operator learning for the placement of a rigid object in a fluid field.

However, despite the high training costs, the DeepONets proved to be accurate for healthcare related problems. The performance of the DeepONets in training, validation, and testing are an order of magnitude more accurate than a random forest baseline. A majority of the validation and testing configurations are sufficiently accurate, with a relative error under 10%. The few validation/test cases that represent the bulk of the error have their errors concentrated in the wake of the surgical light, either by not identifying the wake behavior or mischaracterizing it.

Another consideration is the connection between complex wake behavior and surgical light angle. A more streamlined surgical light may result in less flow separation and thereby avoid the formation of a complex, nonlinear wake. A similar situation occurs when the angle of the surgical light is aligned with the upstream flow such that flow moves over the side of the surgical light. If the angle is changed enough so that air flows over the top of the light, flow separation is more likely to occur. This effect is similar to that of an airflow at low and high angles of attack around objects [35, 36]. Despite this, uniform sampling was used, treating the separated and unseparated flows equally.

This suggests three potential approaches for future work. The first approach is to include boundary conditions inside of PIR and to use turbulence modeling to modify the viscosity near a boundary. In doing so, the loss function will better represent the physics of the problem and may capture the wake behavior more accurately. The second approach is to use mesh refinement to improve the accuracy of the dataset and wake behaviors. The third approach is to use non-uniform or adaptive sampling to obtain a dataset that is more representative of the complexity of the flow behavior [37].

## Acknowledgments

The authors would like to gratefully acknowledge the U.S. Centers for Disease Control and Prevention (CDC) for funding this work. This material is based upon work supported by the Naval Sea Systems Command (NAVSEA) under Contract No. N00024-13-D-6400, Task Orders NH076 and NHP19. Any opinions, findings and conclusions or recommendations expressed in this material are those of the author(s) and do not necessarily reflect the views of the NAVSEA or the U.S. CDC.

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

## A  Related work

SciML has recently shown success when applied to scientific domains [6]. In particular, physics-informed neural networks (PINNs) [38] can solve forward and inverse problems. PINNs have been applied in a wide variety of settings, including fluid flow problems governed by the NSE [39, 40]. PINNs are flexible and can be applied to both uniform and complex geometries, especially when combined with geometric extensions [41]. However, training networks based on PIR has been shown to often be difficult and susceptible to various pathologies [42, 43, 44]. In addition, PINNs solve only a single instance of a PDE, whereas here we focus on the challenge of predicting general PDE solutions.

This latter problem has been solved with a variety of operator-learning models [18, 32, 45, 46, 47, 33, 22, 48], such as the DeepONet [18] and FNO [32], have shown recent success in rapidly predicting solutions to systems governed by nonlinear PDEs, such as the Navier-Stokes equations. However,

these efforts have primarily focused on systems with uniform domain geometries (even though both the DeepONet and an extension of the FNO [49] can be applied to complex geometries); in contrast, many domains of interest in engineering are irregular in some way. Furthermore, many operator-learning approaches focus on predicting solutions to time-dependent systems; here, we consider systems with quasi steady-state dynamics of interest. SciML has been successfully to predict solutions to these systems [50].

Because the interaction of objects with the airflow may create recirculation regions and may potentially allow aerosols to enter the surgical zone, some recent research has questioned the effectiveness of UDF at reducing SSIs [51, 52, 53, 54, 55].

# B Data generation

Simulations were performed using the commercial software COMSOL Multiphysics® version 6.1 [56] to solve for the airflow in a 2D-OR using the incompressible Navier-Stokes equations with the realizable $k$-$\varepsilon$ turbulence model [20]. The velocity at the air supply is set to $35\,\mathrm{ft/min}$ and a pressure outlet is used on both air returns. The 2D-OR measures $27\,\mathrm{ft}$ wide and $10\,\mathrm{ft}$ tall. A $76\,\mathrm{in}\times30\,\mathrm{in}$ rectangular surgical table is centered on the floor of the room with a $70\,\mathrm{in}\times7\,\mathrm{in}$ block placed centered on top of the table to represent the patient. The surgical light is represented as a half-ellipse with semi-axis lengths of $9\,\mathrm{in}$ and $4\,\mathrm{in}$ cut in half along the longer axis. The corners of the resulting shape are then filleted with a radius of $0.05\,\mathrm{ft}$. The air supply is centered on the ceiling and measures $100\,\mathrm{in}$ wide and the air returns were positioned $8\,\mathrm{in}$ off the floor on the left and right wells. Each air return measures $3\,\mathrm{ft}$ tall. See Figure 1 for the OR geometry and an example velocity field.

Steady-state solutions were generated across a uniform discretization of the light position and angle parameter space. The center of the light is positioned within a rectangular region that is $6\,\mathrm{ft}$ wide and $3.2\,\mathrm{ft}$ high, whose center is located $6.4\,\mathrm{ft}$ above the floor and aligned with the middle of the surgical table. The angle of the light ranged from $-90°$ to $90°$ relative to the light pointing directly downwards. The discretization divided the 3-dimensional parameter space into $11\times11\times11$ uniformly-spaced points. Given that COMSOL generates an unstructured mesh based on the specifics of a light's geometry, the number $M_n$ of mesh points for each solution varies. The average and standard deviation number of mesh points per solution was 14106 and 330, respectively. The solutions for 24 (1.8%) of the sampled light orientations did not converge; the orientation of those cases are in Figure 5 in Appendix D. Solutions were generated using a 24-core Intel Broadwell Processor. The average and standard deviation of the compute time per solution was 53.4 CPU-minutes and 25.6 CPU-minutes, respectively.

# C Model hyperparameters

Table 3: Hyperparameters searched across for the RF baseline. The optimal configuration was chosen by evaluation on the validation split, via an exhaustive grid search.

| Hyperparameter | Range of values | Chosen value |
|---|---|---|
| n_estimators | $100, 500$ | 100 |
| min_samples_leaf | $2, 10, 20$ | 2 |
| min_samples_split | $2, 10, 40$ | 2 |

# D Supplemental figures

Table 4: Hyperparameters used for the DeepONet models. The spatial encoder $f_{\text{trunk}}$, parameter encoder $f_{\text{branch}}$, and decoders $f_{(\text{combine},\cdot)}$ (see Eq. 5) were all MLPs with the same architecture. The parameters were chosen by experimentation on the training and validation sets.

| Hyperparameter | Chosen value |
|---|---|
| optimizer | Adam |
| learning rate | $10^{-4}$ |
| # hidden units | 128 |
| # hidden layers | 2 |
| $\lambda_0$ | 1 |
| minibatch size | 4000 |

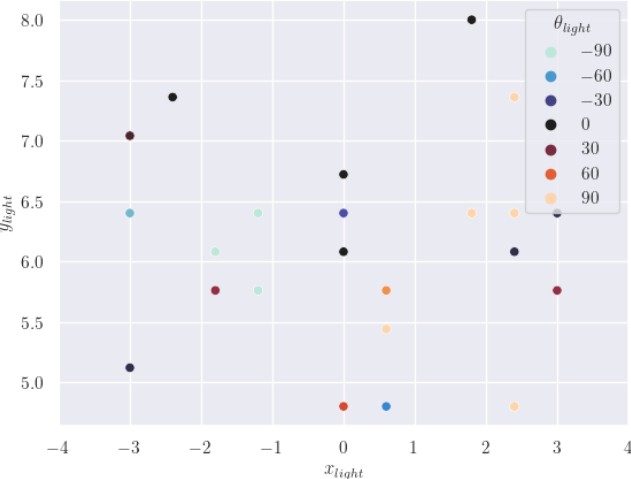

Figure 5: We visualize for what light configurations COMSOL failed to converge, where the point color is the surgical light angle. These light configurations do not represent un-physical simulutions, and they are distributed throughout configuration-space. Further variation of solver options could enable COMSOL to converge.

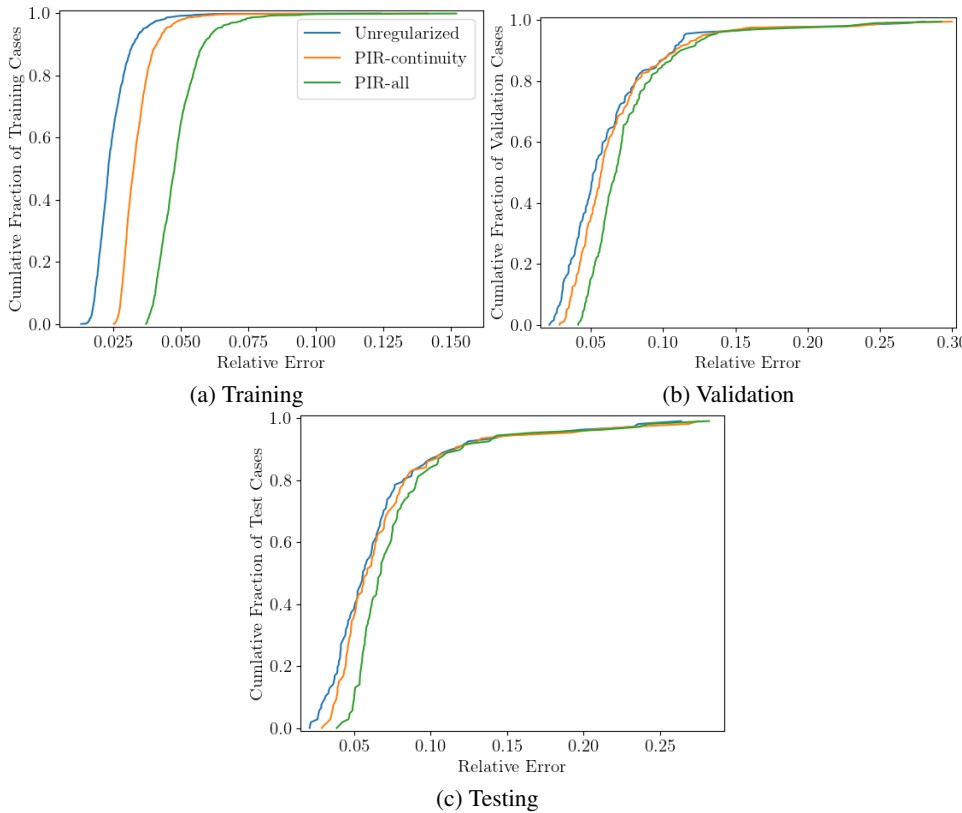

(a) Training  (b) Validation

(c) Testing

Figure 6: On the validation and test split, the unregularized and PIR-continuity models perform comparably, while the PIR-all model experiences some accuracy degradation. We show this using the empirical cumulative distribution function (ECDF) of the relative error (Eq. 12) across each data partition, for the unregularized DeepONet model after 5000 epochs and the PIR-continuity and PIR-all models after an 2000 additional epochs.

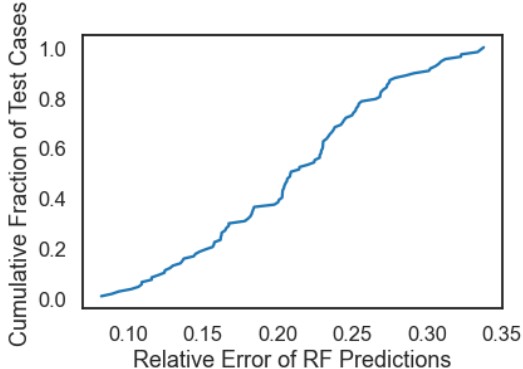

Figure 7: For our RF baseline, we plot the ECDF of the relative errors (Eq. 12) for the test set. The mean relative error is 0.21, which is noticeably higher than the mean relative error for the DeepONet models (6).

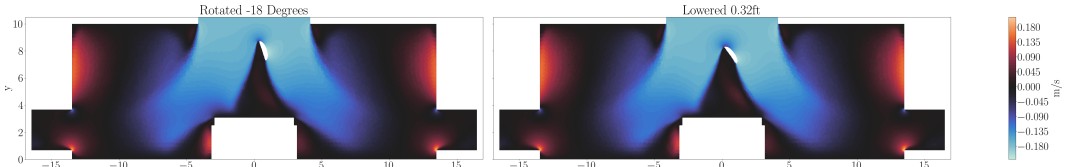

Figure 8: As shown in these vertical velocity profiles, these two configurations from the training set lack strong vortex shedding and reverse flow. These two configurations were the most similar to the validation configuration that had the worst error (Figure 3). Unlike these configurations, the validation configuration did have vortex shedding and reverse flow. Their titles indicate the orientation of the light relative to that of the worst validation case in Figure 3.

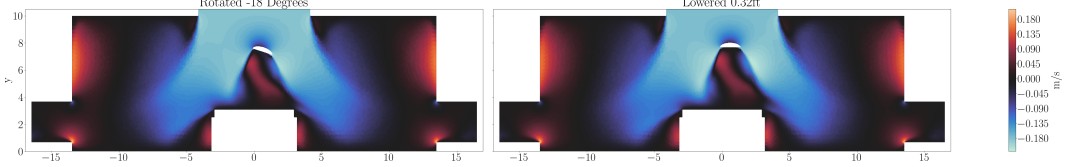

Figure 9: As shown in these vertical velocity profiles, these two configurations from the training set have strong vortex shedding and reverse flow. These two configurations were the most similar to the validation configuration shown in Figure 4, where the models accurately predicted vortex shedding and reverse flow. Their titles indicate the orientation of the light relative to the light orientation in Figure 4.

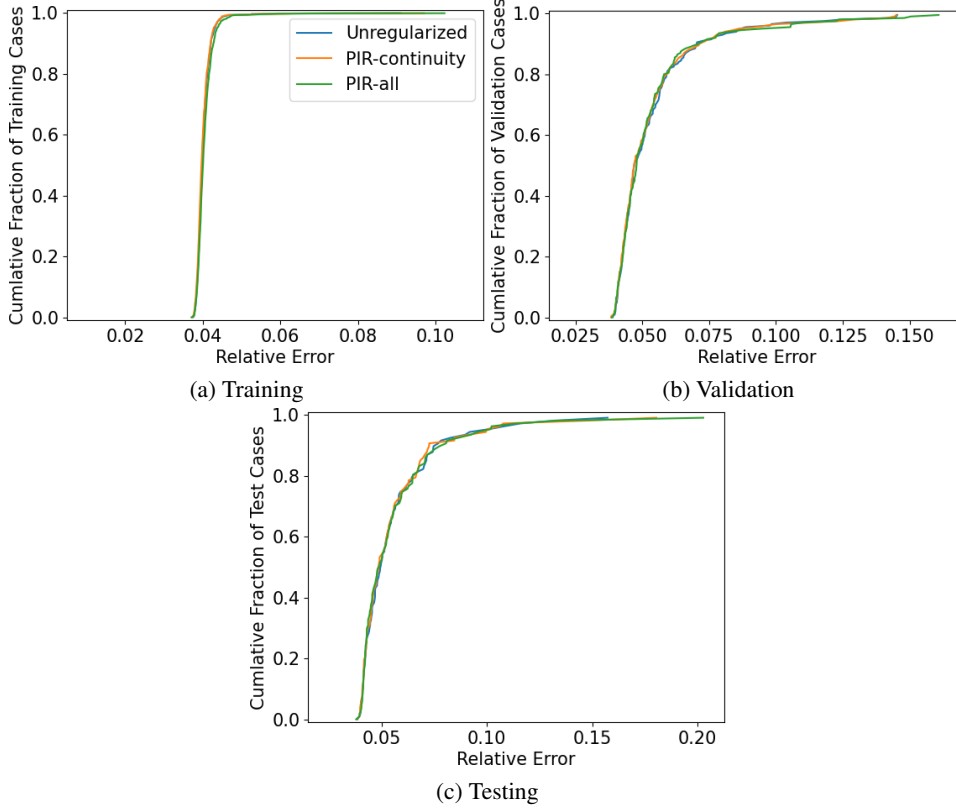

(a) Training

(b) Validation

(c) Testing

Figure 10: When the models are evaluated on a uniform mesh, the differences between each Deep-ONet's accuracy become minimal. This is due to PIR having the strongest impact on predictions in the room geometry that are highly-sampled in the CFD mesh. Evaluation is performed with the ECDF of the relative error (Eq. 12) across each data partition, for each model, evaluated on a regular mesh. The CFD "ground truth" data is obtained from a nearest neighbor approximation of the solution onto a $128 \times 64$ structured mesh and removal of the light, walls, and surgical table geometries. The nearest neighbors approximation was performed with SCIKIT-LEARN's [31] function and with default settings. Errors are computed with Equation (12).

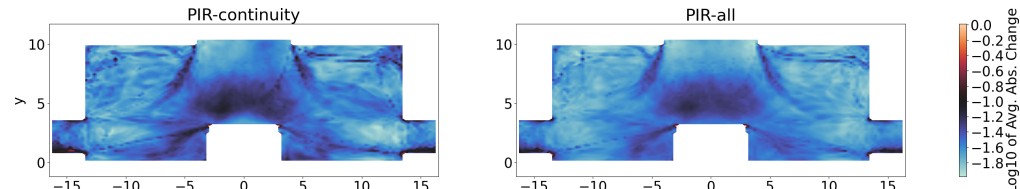

Figure 11: The difference in predicted state variables between PIR and unregularized models is concentrated in the region between the light and the surgical table and near the geometries of the room. This uses the average absolute change (Eq. 13), across all validation and test configurations for the PIR models after 2000 epochs of training, and for the unregularized DeepONet model after 5000 epochs.

