# OpenReview forum: "Rapid Prediction of Two-Dimensional Airflow in an Operating Room using Scientific Machine Learning"
_NeurIPS.cc/2023/Workshop/AI4Science — NeurIPS2023-AI4Science Poster_

### Official Review · Reviewer_Y7LT · 2023-10-21
**Review for paper 13**

**Rating:** 6
**Confidence:** 5

**Review:**

Summary:
The paper proposes the use of DeepONets to model the airflow (x & y-velocities along with pressure) inside an operating room. The paper demonstrates that the DeepONets without physics regularization is able to capture the complex airflow inside the operation room.

Strengths:
1. The application of DeepONets for solving PDEs that govern the airflow inside an operation room is quite interesting and novel.
2. The paper is easy to read and the method is technically sound.

Weaknesses:
1. Random Forest is a very weak baseline to solve PDEs. It would be good to compare against CNNs or even traditional MLP.
2. The x and y-axes labels for the plots (Figure 2) can be more detailed to improve clarity.

---

### Official Review · Reviewer_viNU · 2023-10-24
**Relevant application of DeepONets in the healthcare domain**

**Rating:** 5
**Confidence:** 4

**Review:**

**Strengths**

**S1:** This paper outlines a technically solid application of DeepONets on a simulated counterpart of a highly relevant real-life setting in the healthcare domain. Further development of the work could lead to a useful real-life application of deep operator learning models.

**S2:** The paper does a good job in qualitatively inspecting both failure and success cases of the model, providing insight into the model’s strong and weak points. This type of evaluation is absolutely necessary for such a model to become useful in a sensitive context like healthcare, and can serve as an example to the rest of the community.

**S3:** Physics-informed regularization, as implemented in this method, substantially reduces the physical residuals of the model relative to the purely data-driven, uninformed counterpart. This could prove valuable in demonstrating the degree to which this model is capable of learning the actual physics underlying the data. I encourage the authors to highlight this result more in a revised version of the paper.

---

**Comments**

**C1:** This paper follows the line of research of developing neural PDE solvers that can serve as fast surrogate models for their accurate but numerically expensive counterparts. However, for a model to be useful in this setting, it has to be the case that the benefits of having the fast simulation model outweigh the costs of generating the training set and training the model. This can only be true it the expected number of inference simulations is large. The need for such a large number of simulations in the operating room context has not been sufficiently motivated to justify the large costs of generating >1000 trajectories for training and validation, and the large amount of GPU hours needed for training. In this light, it remains unclear how useful such a model could actually be in a practical setting in this context.

**C2:** The authors mention that PIR does not improve model accuracy and attribute this to misspecification of the physical prior with respect to the data generating process. I have two remarks with respect to this observation. The first is that I would expect that such a physical prior could still be hugely beneficial in the small data regime, and this could aid in avoiding the large amounts of training data needed to train neural PDE surrogates. This remains unexplored in this paper. The second remark is that the authors do observe a huge decrease of the physical residuals, but the presentation of the results mostly stresses that accuracy in terms of relative error did not substantially improve. One could instead argue that relative error (or any other MSE-like error) may not be the right metric for evaluating predictions of turbulent flows, but physical residuals are actually more important.

**C3:** The choice of a random forest baseline is not motivated and rather unusual. Why did the authors not compare to any other neural PDE surrogate baseline method, for example one or more of FNO [1], MPPDE [2], or convolution-based models [3, 4]? This would clarify how the method proposed by the authors compares to models that are commonly used by the community. Such a comparison is necessary to support the claim made in the abstract that 'Existing SciML models struggle in predicting flow when complex geometries determine localized behavior'. The above-mentioned methods can be readily conditioned on the parameterization $\psi$ of the surgical light used in this paper to perform conditional simulation, so they are applicable to this setting.

**C4:** Figure 2 is a bit misleading because it suggests that the regularized models start from untrained weights (epoch 1), while this is not true. To properly interpret the plot, it would help to align the curves on the x-axis so that they start from the epoch after which the unregularized training phase has stopped.

**C5:** For future work, it could be useful to exploit the differentiable nature of the proposed method for inverse design applications, e.g. finding the optimal placement of the light and other attributes in the room so that the probability of pathogens/aerosols contaminating the patient is minimized.

---

**Conclusion**

Overall, the modeling approach of this paper looks technically solid, and further development of the work could lead to an application of scientific machine learning in an impactful domain. However, I have concerns with respect to the motivation of the paper (C1) and the experiments and interpretation of the results (C2 and C3), as well as some minor remarks and suggestions (C4 and C5). Considering that this is a workshop which encourages the submission of work-in-progress papers, C2 is not too problematic in my opinion. However, as the motivation of using neural PDE surrogates in this setting is not clear (see C1) and the experiments do not support some of the claims made in the abstract (see C3), I tend to reject this paper in its current state. Still, I encourage the authors to address these comments and revise the manuscript, and more generally to continue this line of work, which could have a large impact once further developed.

---

**References**

[1] Li, Zongyi, et al. "Fourier neural operator for parametric partial differential equations." arXiv preprint arXiv:2010.08895 (2020).

[2] Brandstetter, Johannes, Daniel Worrall, and Max Welling. "Message passing neural PDE solvers." arXiv preprint arXiv:2202.03376 (2022).

[3] Stachenfeld, Kimberly, et al. "Learned coarse models for efficient turbulence simulation." arXiv preprint arXiv:2112.15275 (2021).

[4] Gupta, Jayesh K., and Johannes Brandstetter. "Towards multi-spatiotemporal-scale generalized pde modeling." arXiv preprint arXiv:2209.15616 (2022).

---

### Meta-Review · Area_Chair_VT7G · 2023-10-27

**Recommendation:** Accept (Poster)
**Confidence:** 3

**Metareview:**

The paper discusses the application of DeepONets to model airflow inside an operating room. The reviewers have highlighted several strengths and weaknesses of the paper, along with some suggestions and comments.

Considering the strengths and weaknesses outlined by the reviewers, as well as the comments and suggestions, it appears that the paper has potential but requires revisions and further clarification. In its current state, there are concerns about motivation, baseline comparison, and the choice of evaluation metrics.

The paper could benefit from a more comprehensive comparison with existing neural PDE surrogate methods and a clearer justification for its practical utility in healthcare settings.